# Hepatoprotective Effects of Flavonoids against Benzo[a]Pyrene-Induced Oxidative Liver Damage along Its Metabolic Pathways

**DOI:** 10.3390/antiox13020180

**Published:** 2024-01-31

**Authors:** Min Kim, Seung-Cheol Jee, Jung-Suk Sung

**Affiliations:** Department of Life Science, Dongguk University-Seoul, Goyang 10326, Republic of Korea; pipikimmin@dongguk.edu (M.K.); markjee@dongguk.edu (S.-C.J.)

**Keywords:** benzo[a]pyrene, flavonoids, liver damage

## Abstract

Benzo[a]pyrene (B[a]P), a highly carcinogenic polycyclic aromatic hydrocarbon primarily formed during incomplete organic matter combustion, undergoes a series of hepatic metabolic reactions once absorbed into the body. B[a]P contributes to liver damage, ranging from molecular DNA damage to the onset and progression of various diseases, including cancer. Specifically, B[a]P induces oxidative stress via reactive oxygen species generation within cells. Consequently, more research has focused on exploring the underlying mechanisms of B[a]P-induced oxidative stress and potential strategies to counter its hepatic toxicity. Flavonoids, natural compounds abundant in plants and renowned for their antioxidant properties, possess the ability to neutralize the adverse effects of free radicals effectively. Although extensive research has investigated the antioxidant effects of flavonoids, limited research has delved into their potential in regulating B[a]P metabolism to alleviate oxidative stress. This review aims to consolidate current knowledge on B[a]P-induced liver oxidative stress and examines the role of flavonoids in mitigating its toxicity.

## 1. Introduction

Benzo[a]pyrene (B[a]P) is a prevalent environmental pollutant present in tobacco smoke, vehicle emissions, and industrial processes, posing serious health risks to human health and the environment [1]. Upon absorption, B[a]P primarily undergoes hepatic metabolic transformations through the actions of enzymes, such as CYP1A1 and CYP1B1, to form B[a]P-7,8-epoxide [2]. Further metabolic processing involving epoxide hydrolase leads to the formation of B[a]P-7,8-dihydrodiol, which is subsequently converted by CYP1 enzymes into B[a]P-7,8-diol-9,10-epoxide (BPDE), a highly reactive intermediate [3].

BPDE, identified as a potent carcinogen, induces DNA damage by forming covalent adducts upon interaction with DNA molecules, thereby leading to mutations and initiating tumorigenesis [1]. The liver, central in B[a]P metabolism, is particularly vulnerable to its toxic effects [4]. Liver tissues have exhibited BPDE-DNA adducts, which are associated with liver cancer development [5]. Additionally, BPDE can induce oxidative stress, leading to an increase in reactive oxygen species (ROS) production [6]. ROS can further damage cellular components, including DNA, proteins, and lipids, contributing to cellular dysfunction and inflammation [7]. This oxidative stress cascade can exacerbate the toxicity of B[a]P and its metabolites, resulting in a spectrum of adverse health effects [8].

Minimizing human exposure to B[a]P and controlling environmental pollution are imperative due to the severe health implications associated with its exposure. Understanding the metabolic and toxic impacts of B[a]P is pivotal in developing preventive and therapeutic measures aimed at safeguarding human liver health. The liver is an important organ that is involved in various metabolic activities. Chronic hepatic damage leads to conditions such as cirrhosis, fibrosis, fatty liver, and even hepatocellular carcinoma, raising serious epidemiological concerns [9]. Despite remarkable progress in modern medicine, the lack of suitable and effective hepatoprotective drugs remains a persistent concern. The withdrawal of numerous drugs from the market due to drug-induced liver injury (DILI) has prompted research into alternative treatment approaches [10]. Currently, the use of naturally derived phytochemicals to treat a variety of diseases is becoming increasingly accepted. Among these bioactive compounds, flavonoids are well-known for their natural origin. Ongoing research into flavonoids has revealed promising health benefits, particularly their reported positive effects on human health [11]. Considering the hepatic metabolism of xenobiotic flavonoids, a comprehensive review of their specific impacts on liver health is essential. Especially, the single or synergistic effect of flavonoids against B[a]P-induced toxicity and drug metabolism have been studied. By exploring their mechanisms of action and interactions within the liver, this review aims to understand the role of flavonoids in promoting liver health and countering B[a]P-induced oxidative stress. In Section 2, we primarily aimed to communicate about oxidative damage during B[a]P metabolism and the subsequent liver damage. Then, in Section 3, the focus shifts to exploring flavonoids as a potential solution to mitigate this damage.

## 2. B[a]P-Induced Oxidative Liver Damage

### 2.1. Oxidative Damage Induced by B[a]P in the Liver

Polycyclic aromatic hydrocarbons (PAHs) undergo metabolic transformation via xenobiotic-metabolizing enzymes upon body absorption. Among these, B[a]P, a PAH, engages in a carcinogenic activation process through three distinct pathways: (1) Radical-cation; (2) Diol epoxide; (3) Quinone pathways [3,12]. In the radical-cation pathway, hydroxylated metabolites, including B[a]P-quinones, such as B[a]P-1,6-dione and B[a]P-6,12-dione, are produced. These B[a]P-quinones, highly reactive due to their chemical structure, undergo enzymatic reduction by NAD(P)H quinone dehydrogenase 1 (NQO1), forming hydroquinone equivalents. Generated from B[a]P-quinones, hydroquinones swiftly undergo two autooxidation cycles, yielding radicals while regenerating quinones. Consequently, these futile cycles result in the combination of molecular oxygen (O_2_) molecules with B[a]P metabolites, generating superoxide anion radicals (O_2_^−^) and hydrogen peroxide (H_2_O_2_) [13]. The metabolic processing of B[a]P inevitably leads to the production of ROS (Figure 1).

ROS function as critical signaling molecules that determine cell fate; however, their excessive accumulation can lead to irreversible cellular damage or death [14]. ROS is a highly reactive molecule that attacks DNA [7]. Particularly, guanine, with the lowest ionization potential among nucleic acid components, becomes a preferred target for one-electron oxidizers, either directly or indirectly [15]. Consequently, the predominant manifestation of oxidative DNA damage is 8-oxo-7,8-dihydro-2′-deoxyguanosine (8-oxo-dG), generated through the oxidation of the reducing purine radical. Furthermore, the side-oxidation of 8-oxo-dG by highly oxidizing oxyl radicals may lead to the synthesis of oxazolone and imidazolone nucleosides [16]. B[a]P induces ROS release, promoting genotoxicity by generating 8-oxo-dG [17]. In mice orally administered 2 mg/kg of B[a]P for 55 days, a significant increase in 8-oxo-dG levels occurred in the liver, stomach, colon, and kidney by more than a 2–3 fold change. Therefore, 8-oxo-dG is a common form of oxidative DNA damage and serves as a prominent marker of oxidative DNA damage within the DNA molecule [18]. This type of DNA damage can cause mutations in the genetic material, DNA by influencing DNA replication and DNA repair systems, potentially leading to the accumulation of DNA damage and mutations [19]. These mutations can disrupt basic cellular functions such as cell cycle control, cell division, DNA replication, and DNA repair, often associated with cancer development. Specifically, 8-oxo-dG is an important indicator of DNA damage, with numerous studies reporting its relevance to cancer [20,21,22]. The increased presence of 8-oxo-dG correlates with cancer initiation and progression, potentially enhancing carcinogenicity [23]. This accumulation of DNA damage and mutations, with 8-oxo-dG as a key contributor, can significantly contribute to cancer development, positioning it as a crucial biological marker associated with cancer initiation and progression.

In addition to DNA damage, B[a]P-induced ROS triggers various cellular dysfunctions, particularly impacting mitochondrial DNA, which constitutes approximately 1% of total cellular DNA and is believed to be highly susceptible to oxidative stress [24]. Persistent damage to mitochondrial DNA eventually leads to mutations within the mitochondrial genome, exacerbating mitochondrial dysfunction and contributing to the onset and progression of diverse diseases. Moreover, studies reveal a dose-dependent increase in mitochondrial depletion following B[a]P exposure, particularly impacting the liver more significantly than other organs [25]. This emphasizes the significant influence of B[a]P on mitochondrial dysfunction and its potential implications for overall physiological pathogenesis. Mitochondrial dysfunctions disrupt the electron flow in the electron transport chain (ETC), causing electron leakage onto O_2_, thereby elevating ROS production within damaged mitochondria. Altered ROS levels linked to mitochondrial activity result from disruptions in mitochondrial metabolic processes, such as NADH accumulation in dysfunctional mitochondria. Additionally, damaged mitochondria indirectly elevate ROS production from the endoplasmic reticulum surface [26]. These damaged mitochondria can prematurely release electrons, particularly from Complex I (NADH:ubiquinone oxidoreductase) or Complex II (succinate dehydrogenase) within the ETC, reacting with O_2_ to produce superoxide radicals (O_2_^•−^)—a primary source of ROS [27]. Moreover, damaged mitochondria exhibit impairments in electron carriers and ETC proteins, rendering them more susceptible to electron leakage and subsequent ROS generation [28]. The ROS produced within damaged mitochondria then contribute to a “vicious cycle” by causing additional damage to mitochondrial components, such as lipids, proteins, and nucleic acids (including mtDNA). This cycle of mitochondrial dysfunction and ROS production culminates in further cellular damage and impairments in cellular function.

The response to B[a]P-induced ROS damage varies across different organs [29]. Acute B[a]P exposure significantly decreased body weight across various doses. Previous research indicates variations in oxidative DNA damage among organ tissues, with a prevalence sequence as follows: liver > lung > kidney > brain > stomach [29]. Moreover, B[a]P induces relatively higher oxidative stress in the liver and lungs but exhibits less oxidative stress in the stomach and brain. The liver is one of the essential organs in xenobiotic metabolism and is predominantly involved in B[a]P metabolism. Chronic liver diseases consistently exhibit increased oxidative stress, regardless of their origin. ROS plays a pivotal role in the initiation and progression of numerous liver disorders. Additionally, oxidative stress influences liver fibrogenesis [30].

Exploring the interactions between B[a]P and ROS yields valuable insights into its potential carcinogenicity, cellular dysfunctions, and diverse organ impacts, particularly the liver. Understanding these mechanisms is essential for formulating effective strategies to alleviate the adverse effects of B[a]P and improve B[a]P-induced liver diseases.

### 2.2. Metabolism and Excretion of B[a]P and Their Hepatic Consequences 

The liver exhibits heightened sensitivity to B[a]P compared with other organs due to its central role as the primary site for drug metabolism (Figure 2). As the primary organ responsible for xenobiotic drug metabolism, the liver significantly contributes to minimizing drug toxicity. This organ undertakes a multi-phase process in xenobiotic metabolism. Phase I involves cytochrome P450 oxidases catalyzing the oxidation or reduction in xenobiotic materials. In phase II, these modified xenobiotics are conjugated with other polar compounds by specific phase II enzymes. Finally, phase III involves the excretion of conjugated xenobiotic materials from cells through efflux transporters, effectively eliminating them from the cells [31].

B[a]P undergoes a sequence of metabolic transformations catalyzed primarily by CYP1 enzymes, such as CYP1A1 and CYP1B1. Initially, B[a]P is metabolized into B[a]P-7,8-epoxide, which is subsequently modified via hydrolysis by epoxide hydrolase, resulting in the formation of B[a]P-7,8-dihydrodiol. Further enzymatic action by CYP1 converts B[a]P-7,8-dihydrodiol into BPDE, a highly reactive compound [32]. BPDE, a major metabolite, forms covalent bonds with DNA by intercalating and reacting with nucleophilic guanine bases, binding specifically to guanine at the N2 position with multiple alignments within the helix [33]. This process results in the formation of bulky guanine adducts [34]. Specifically, the (K)-7S,8R,9R,10S+anti-B(a)PDE enantiomer (10S) DNA adduct positions within the minor groove, oriented towards the 5′ end of the helix, affecting neighboring bases around the guanine adduct [35]. Consequently, these BPDE DNA adducts induce distortions in DNA structure, leading to molecular modifications [36]. Such genetic alterations can impede proper DNA repair mechanisms. Failure in repair processes may result in the transmission of these DNA lesions to daughter cells during cell division, resulting in the accumulation of DNA damage and potentially contributing to cancer development [37]. Clearly, B[a]P metabolites, particularly BPDE, play a crucial role in inducing genetic damage, which, if left unrepaired, can lead to cancer progression. The crucial involvement of CYP1 enzymes and epoxide hydrolase in B[a]P metabolism to form BPDE cannot be overstated. CYP1 enzymes and epoxide hydrolase are regulated by a common transcription factor, the aryl hydrocarbon receptor (AhR) [38]. B[a]P, functioning as an AhR ligand, activates the receptor, subsequently inducing the expression of downstream genes such as CYP1 enzymes and epoxide hydrolase [39]. In essence, B[a]P, by activating AhR, elevates the expression levels of CYP1 enzymes and epoxide hydrolase, thereby increasing the production of carcinogenic BPDE molecules. This mechanism underscores the significance of AhR signaling in B[a]P-induced carcinogenesis and the potential for liver cancer development. 

Phase II enzymes play a pivotal role in xenobiotic drug metabolism by converting endogenous substances and foreign compounds into more easily excretable forms. This involves processes such as conjugation with glutathione, glucuronidation, or glycine, contributing to the metabolic deactivation of pharmacologically active substances. Glutathione S-transferases (GST), part of the phase II enzyme group, catalyze the conjugation of reduced glutathione (GSH) with xenobiotic substrates, a crucial step in detoxification [40]. The conjugation of BPDE with GSH represents a pivotal mechanism in cellular detoxification [41]. Another essential superfamily within phase II metabolic enzymes, uridine diphosphate glucuronosyltransferases (UGTs), facilitate the transfer of glucuronosyl groups from uridine 5′-diphosphate-glucuronic acid to substrates containing alcohols, amines, or carboxylic acids as functional groups [42]. Notably, key UGT isoforms such as UGT1A3 and 2B7 have been found to inhibit B[a]P-7,8 catechol. Consequently, decreased phase II enzyme metabolism capacity can lead to the accumulation of xenobiotic material toxicity [43]. Hence, maintaining the activity and expression of phase II enzymes is critical in mitigating the toxic effects of B[a]P. However, research suggests that B[a]P may disrupt the normal regulation of phase II enzymes. For instance, in mice treated with B[a]P, GST activity showed no significant changes in the liver compared with the non-treated group, contrary to the expected activation of GST by typical phase II activation pathways [44]. Typically, phase II enzymes are effectively activated to detoxify xenobiotics formed by phase I enzymes. This discrepancy indicates that the increased BPDE produced in the activated phase I step might pose challenges for proper detoxification. Several studies reveal varying activity and expression levels of GST depending on specific enzyme types and organ types, suggesting diverse susceptibility to B[a]P. Notably, in the lungs, B[a]P significantly induces total GST expression, although they are not activated by B[a]P in the lung [44]. Recent findings revealed the capacity of B[a]P to enhance the activity of specific GST enzymes such as GSTA2, M1, and P1 while concurrently reducing GSTA4 activity [45]. This highlights the crucial need for precise regulation of phase II enzymes inhibited by benzo[a]pyrene to effectively mitigate its toxic impact.

Phase III enzymes, including a diverse group of membrane transporters such as the multidrug resistance protein family [16], aid in the elimination of conjugated metabolites from cells. These proteins, belonging to the ATP-binding cassette (ABC) transporters, facilitate ATP-dependent transport of various hydrophobic anions [46]. Consequently, ABC transporters play a critical role in expelling phase II metabolites out of cells, enabling further metabolism or excretion to occur [47]. The complex detoxification response of ABC transporters to B[a]P exposure warrants further investigation due to limited available information. Nevertheless, some studies suggest their involvement in B[a]P excretion. For instance, in various species, ABC transporters have been identified as responsible for eliminating B[a]P metabolites [48,49]. Furthermore, B[a]P has been shown to reduce the expression levels of ABCA12 [50] and decrease the protein expression level of ABCC1 [51], suggesting a potential hindrance in the proper removal of detoxified B[a]P metabolites from cells due to reduced transporter expression.

In summary, exploring the regulation of genes associated with the xenobiotic metabolism B[a]P will provide valuable insights into the susceptibility of the liver to B[a]P. 

## 3. Hepatoprotective Potential of Flavonoids against B[a]P-Induced Liver Damage

### 3.1. Classification of Flavonoids

Flavonoids, a subgroup of polyphenols, represent plant-derived compounds renowned for their antioxidant properties [52]. These low-molecular-weight compounds are secondary metabolites found in plants [53], playing significant roles by contributing to color, aroma, and important responses to biological and non-biological environmental factors, such as their growth and defense against pathogens and UV radiation [54,55]. Consequently, flavonoids are ubiquitous across plant species. Chemically, flavonoids feature a 15-carbon backbone structure composed of two benzene rings linked by a heterocyclic pyran ring [56]. Hence, they are denoted as C6-C3-C6 compounds (Figure 3).

Flavonoids, distinguished by their diverse chemical structures and properties, are categorized into several groups based on their chemical structure, oxidation degree, and unsaturation in the linking chain (C3). These groups encompass anthocyanidins, chalcones, flavonols, flavanones, flavanonols, flavones, and isoflavonoids. Anthocyanidins are a group of colorful pigments in plants responsible for vibrant colors ranging from red to blue. Key types include pelargonidin, cyanidin, peonidin, delphinidin, malvidin, and petunidin [57]. Chalcones are a class of compounds that serve as precursors to flavonoids, including marene, phloretin, lycochalcone-A, isobachalcone, and xanthohumol [58,59]. Flavonols are a subgroup of flavonoids characterized by the 3-hydroxyflavone backbone, comprising quercetin, kaempferol, rutin, myricetin, isoquercetin, and isorhamnetin [60]. Flavanones are various aromatic, colorless ketones derived from flavone, commonly occurring as glycosides in plants. Examples include naringenin, hesperidin, hesperetin, eriodictyol, and sakuranetin [61]. Flavanonols are a group of flavonoids that use the 3-hydroxy-2,3-dihydro-2-phenylchromen-4-one backbone [62]. This category includes taxifolin, aromadendrin, and engeletin. Flavones are a class of flavonoids based on the 2-phenylchromen-4-one backbone [63]. Well-known flavones include apigenin, wogonin, luteolin, baicalein, tangeretin, and chrysin [64]. While flavonoids have the 2-phenylchromen-4-one backbone, isoflavonoids, such as genistein and daidzein, feature the 3-phenylchromen-4-one backbone without hydroxyl group substitution at position 2 (isoflavones) or the 3-phenylchroman (isoflavan) backbone [65]. These varied flavonoid classes are abundant in various plant-based foods and have garnered significant scientific interest due to their potential health benefits and bioactive properties. 

Considering that flavonoids, being xenobiotics, undergo metabolism in the liver, it becomes crucial to explore their specific impacts on liver health comprehensively. The following sections aim to delve into their mechanisms of action and interactions within the liver to enhance our understanding of how flavonoids may alleviate benzopyrene-induced liver damage and aid in maintaining optimal liver health.

### 3.2. Antioxidant Effects of Flavonoids and B[a]P in the Liver

Oxidative stress is a pivotal factor contributing to various liver diseases, including drug-induced liver injury, viral hepatitis, and alcoholic hepatitis. These conditions are triggered by the detrimental effects of drugs, viruses, and alcohol on the liver [66], thereby underscoring the significance of understanding and addressing the mechanisms underlying oxidative stress to prevent and manage these diverse liver conditions. Flavonoids are widely recognized as antioxidant compounds, prompting extensive research into their antioxidant potential [52]. Their antioxidant effects are achieved through two main mechanisms. First, they involve a chemical radical-scavenging effect that neutralizes radicals [67]. Second, they regulate the complex system of antioxidant enzymes, serving as a defense mechanism against intracellular ROS. Regarding the radical scavenging activity of flavonoids, previous reports have highlighted the influence of superoxide anion scavenging on their antioxidant properties [68]. However, a more recent study has introduced exceptions to the correlation between the antioxidant effect and scavenging activities of flavonoids [69]. This study explored the superoxide anion-scavenging potency and antioxidation capacity of seven specific flavonoids—quercetin, rutin, morin, acacetin, hispidulin, hesperidin, and naringin—and revealed distinct variations in scavenging ability. Rutin emerged as the most potent scavenger, closely followed by quercetin. Conversely, naringin and hesperidin exhibited no antioxidative effects. Furthermore, other research demonstrated variations in the roles of flavonoids, indicating that while flavonoids with scavenging effects exhibit antioxidant activity, those lacking such effects also possess antioxidant properties. This suggests that radical-scavenging properties are not universal among all flavonoids. Rather, not all flavonoids have radical-scavenging properties; different flavonoids exhibit antioxidant effects through different mechanisms. 

Flavonoids act as a defense mechanism against intracellular ROS by regulating a complex system of antioxidant enzymes. Numerous studies have explored the nuclear factor E2-related factor 2 (NRF2)-HO antioxidant enzyme pathway (Table 1). Nrf2 serves as a transcription factor responsive to oxidative stress, binding to the antioxidant response element in the promoters of genes encoding antioxidant enzymes [70]. Under homeostatic conditions, KEAP1 cooperates with an E3 ubiquitin ligase to tightly regulate the activity of the transcription factor NRF2 activity through ubiquitination and subsequent proteasome-dependent degradation. In response to stress responses, complex molecular mechanisms involving sensor cysteines within KEAP1 prevent NRF2 ubiquitination, allowing its intracellular accumulation and translocation to the nucleus. There, NRF2 promotes the transcription of antioxidant enzymes such as NQO1, heme oxygenase 1 (HO1), catalase (CAT), superoxide dismutase (SOD), and glutathione peroxidase (GPX) [71,72] all of which function as reductases with the capability to diminish ROS and reduce oxidative stress induced by ROS. SOD acts as a CAT, converting superoxide into O_2_) [73]. CAT catalyzes the breakdown of hydrogen peroxide (H_2_O_2_). HO1 catalyzes heme breakdown. GPX exhibits peroxide activity, protecting against oxidative damage and acting as a reductase to convert H_2_O_2_ to water. NQO1 acts as a detoxification system, removing quinone compounds from the biological environment. Notably, B[a]P triggers ROS generation through redox cycles involving benzoquinones and benzohydroquinones. In this context, the clearance of quinones becomes crucial, and NQO1 might potentially mitigate B[a]P-induced ROS. 

The NRF2 pathway is intricately regulated by various signaling pathways involving transcriptional control, post-translational modifications, and maintenance of NRF2 protein stability [74]. As previously mentioned, Keap1 regulates the NRF2 protein stability, while post-translational modifications of NRF2 influence its binding partners and cellular localization, impacting its stability [75]. Several kinases, including ERK, JNK, PI3K-AKT, and PKC, contribute to NRF2 phosphorylation, enhancing its stability and subsequent transcriptional activity [76]. Conversely, phosphorylation mediated by p38 and GSK3 leads to reduced NRF2 stability. The NFE2L2 gene integrates a positive feedback mechanism, amplifying NRF2 effects [74]. Moreover, NFE2L2 transcription is regulated by various transcription factors, such as AhR and NFkB, with the NFE2L2 promoter containing an NFkB-binding site, allowing regulation by this transcription factor [75].

Numerous studies have demonstrated the regulatory role of flavonoids in modulating NRF2 via diverse mechanisms, leading to reduced ROS levels in the liver. Table 1 provides a comprehensive overview of the hepatic antioxidant mechanisms mediated by flavonoids. These compounds effectively diminish ROS and malondialdehyde, a marker reflecting oxidative stress levels (Table 1). Considering the antioxidant mechanisms of flavonoids in the liver, it is evident that they possess the potential to mitigate the hepatotoxic effects induced by B[a]P or counteract the detrimental impact of ROS-induced liver damage.

**Table 1 antioxidants-13-00180-t001:** Antioxidant properties and the mechanism of flavonoids in the liver.

Flavonoid		ROS	MDA	Antioxidant Enzyme							NRF2 Regulator	Refs.
				NRF2	HO1	NQO	SOD	CAT	GSH	GPX		
Anthocyanidin	Pelargonidin	↓	↓	↑			↑		↑			[77]
	Cyanidin	↓	↓	↑	↑	↑	↑	↑	↑			[78]
		↓					↑		↑		JNK↓	[79]
	Delphinidin	↓		↑	↑	↑					Keap1↓	[80]
		↓					↑				PI3K/Akt↑	[81]
					↑		↑					[82]
Chalcone	Phloretin	↓		↑	↑		↑	↑		↑	AMPK↑ Keap1↓	[83]
		↓	↓	↑	↑		↑			↑	AMPK↑	[84]
		↓		↑	↑				↑		ERK↑	[85]
	Xanthohumol	↓	↓	↑	↑	↑			↑		AKT↑ AMPK↑	[86]
		↓	↓				↑	↑	↑			[87]
Flavonols	Quercetin	↓	↓	↑		↑	↑	↑	↑		Keap1↓	[88]
		↓	↓						↑		NF-kB↓	[89]
		↓	↓	↑	↑				↑		NF-kB↓	[90]
	Kaempferol	↓	↓	↑			↑		↑	↑		[91]
		↓	↓	↑	↑		↑		↑		NF-kB↓	[92]
		↓	↓	↑	↑				↑		AMPK↑	[93]
		↓		↑	↑	↑	↑	↑	↑		AKT↑	[94]
			↓	↑	↑	↑	↑		↑		NF-kB↓	[95]
	Rutin	↓					↑			↑		[96]
		↓					↑	↑		↑		[97]
		↓	↓	↑	↑		↑	↑		↑		[98]
	Myricetin	↓	↓	↑			↑	↑	↑	↑	NF-kB↓	[99]
		↓	↓				↑		↑			[100]
		↓	↓	↑	↑		↑		↑		AMPK↑	[101]
	Isorhamnetin	↓							↑			[102]
Flavanone	Naringenin		↓				↑	↑	↑	↑	NF-kB↓	[103]
		↓	↓				↑	↑	↑	↑	SIRT1↑	[104]
		↓		↑			↑	↑	↑	↑	NF-kB↓	[105]
	Hesperidin	↓	↓	↑			↑	↑	↑		Keap1↓	[106]
		↓	↓	↑	↑		↑		↑	↑	NF-kB↓	[107]
		↓	↓	↑	↑			↑	↑		NF-kB↓	[108]
		↓		↑	↑						MAPK↑	[109]
	Eriodictyol	↓	↓	↑	↑		↑	↑		↑		[110]
Flavanonols	Taxifolin	↓	↓					↑	↑		NF-kB↓	[111]
Flavones	Apigenin	↓	↓	↑	↑	↑	↑	↑	↑		NF-kB↓	[112]
		↓	↓				↑		↑		NF-kB↓	[113]
	Wogonin	↓	↓	↑	↑	↑	↑				NF-kB↓	[114]
	Luteolin		↓	↑	↑	↑	↑				NF-kB↓	[115]
		↓	↓	↑	↑	↑	↑	↑	↑	↑	KEAP↓	[116]
		↓				↑						[117]
	Tangeretin	↓		↑	↑	↑			↑		MAPK↑	[118]
	Chrysin	↓	↓				↑	↑	↑	↑		[119]
Isoflavonoids	Genistein	↓		↑			↑					[120]
	Daidzein	↓						↑				[121]
		↓	↓				↑	↑	↑	↑		[122]

ROS: reactive oxygen species, MDA: malondialdehyde, NRF2: nuclear factor E2-related factor 2, HO1: heme oxygenase 1, SOD: superoxide dismutase, GSH: glutathione, GPX: glutathione peroxidase, NQO: NAD(P)H quinone oxidoreductase 1, CAT: catalase, ↑; Upregulation of gene expression, ↓; Downregulation of gene expression.

### 3.3. Regulation of B[a]P Metabolism by Flavonoids in the Liver

The mitigation of B[a]P toxicity fundamentally relies on the xenobiotic detoxification process [123]. As previously discussed, B[a]P treatment can upregulate phase I CYP enzymes while inhibiting phase II enzymes. In such cases, the phase II step acts as the rate-limiting step, ultimately suppressing detoxification. Consequently, intracellular accumulation of the carcinogenic compound BPDE and the generation of ROS precursor, such as B[a]P quinone, occur, amplifying the overall toxicity. Therefore, precise regulation of the rate and activity of phases I, II, and III is crucial in reducing B[a]P toxicity.

Studies suggest that flavonoids exert regulatory effects on these phase enzymes [124]. Reduced activity in phase I enzymes results in reduced B[a]P metabolism, thereby inhibiting the formation of the carcinogenic compound BPDE [125]. Conversely, increased phase II enzyme activity accelerates the detoxification of substances such as BPDE and B[a]P-quinones [51]. Elevated phase III enzyme activity facilitates the swift excretion of detoxified compounds from the cell to the external environment. When both phase I and phase II enzyme activities increase, the metabolism and detoxification processes accelerate, expediting detoxification mechanisms.

Table 2 provides a comprehensive overview of flavonoid-mediated regulation of phase enzymes and their influence on B[a]P metabolism. For instance, quercetin demonstrates the ability to enhance the expression of all three phases: I, II, and III [51]. This results in a significant enhancement of B[a]P metabolism, facilitating detoxification and efficient excretion of harmful BPDE substances from within the cell. In contrast, kaempferol appears to decelerate B[a]P metabolism and subsequent detoxification [126]. Myricetin, while inhibiting the expression of phase I enzymes, promotes detoxification through phase II [108]. This dual action not only suppresses the production of the carcinogenic BPDE in phase I but also aids in its detoxification during phase II. Similarly, quercetin-like flavonol isorhamnetin accelerates the expression of phase I, II, and III enzymes, resulting in expedited metabolism, detoxification, and excretion [51]. On the other hand, flavones, such as apigenin, wogonin, and luteolin, inhibit the expression of phase I enzymes, reducing the formation of carcinogenic BPDE [126,127,128,129]. In contrast, chrysin enhances the expression of both phase I and II enzymes, suggesting a rapid metabolism and detoxification process [130]. Notably, these flavonols, including quercetin, kaempferol, isorhamnetin, and myricetin, each exhibit distinct mechanisms for mitigating B[a]P toxicity within their shared category.

Furthermore, flavonoids intricately regulate various CYP enzymes, each exerting distinct regulatory effects. Among these, certain CYPs hold pivotal roles in controlling B[a]P metabolism, while others contribute modestly. Through an analysis of B[a]P metabolites generated during incubation with different CYP types, one study revealed that CYP1A1 and CYP1B1 significantly influence B[a]P metabolism [131]. Additionally, essential CYPs involved in B[a]P metabolism in the liver, such as CYP3A4 and CYP2C19, were identified. Moreover, several other CYPs exhibited minor involvement in B[a]P metabolism. Therefore, Table 2 details the regulation of CYPs by flavonoids, categorized according to the type of flavonoid.

Mitigating the toxicity of B[a]P relies heavily on modulating xenobiotic detoxification processes. Flavonoid treatments can significantly influence the expression of phase I, II, and III enzymes, with phase III enzymes often acting as the rate-limiting step in detoxification. Precise regulation of the rates and activities of all three phases is pivotal to reducing B[a]P toxicity effectively. Flavonoids’ regulation of the detoxification metabolic process directly affects the regulation of carcinogenic compounds such as BPDE and ROS precursors. Table 2 provides a comprehensive overview of how flavonoids influence phase enzymes and CYPs, demonstrating their potential to alleviate B[a]P toxicity through distinct mechanisms. The varied actions of flavonoids within the same category highlight the complexity and adaptability of these compounds in addressing B[a]P-induced liver problems.

**Table 2 antioxidants-13-00180-t002:** Effect of flavonoids on xenobiotic metabolism enzymes in the liver.

Flavonoid		Phase I Enzyme				Phase II Enzyme				Phase III Enzyme	B[a]P		Refs.
		AhR	CYP1A1	CYP1B1	Other CYPs	NRF2	PXR	GST	UGT		Metabolism	Detoxification	
Anthocyanidin	Pelargonidin	↑	↑		CYP1A2↑								[132]
Chalcone	Phloretin				CYP1A2↓ CYP3A4↓	↑		↑					[133]
								↑					[134]
Flavonols	Quercetin		↑										[135]
			↑		CYP2↑ CYP3↑								[136]
		↑	↑	↑		↑	↑	↑		ABCC1↑	↑	↑	[51]
	Kaempferol		↓	↓	CYP1A2↓								[137]
					CYP2E1↓								[138]
		↓	↓			↓	↓				↓	↓	[126]
	Rutin		↑					↑					[139]
			↑		CYP1A2↑								[140]
					CYP2E1↑			↑					[141]
	Myricetin				CYP2C8↓								[142]
			↓					↑			↓	↑	[108]
	Isorhamnetin	↑	↑	↑		↑	↑	↑		ABCC1↑	↑	↑	[51]
Flavanone	Eriodictyol				CYP2E1↓ CYP3A11↓			↑	↑				[143]
	Sakuranetin				CYP1A2↓ CYP2C9↓	↑			↓				[144]
Flavanonols	Taxifolin				CYP2E1↓ CYP1A2↓ CYP3A4↓			↑					[145]
Flavones	Apigenin				CYP4F2↓								[146]
		↓	↓	↓	Total contents↓			↑			↓	↑	[127]
	Wogonin		↓								↓		[129]
			↓		CYP2E1↓			n.s.			↓	n.s.	[128]
	Luteolin				CYP1A2↓ CYP3A4↓				↓				[147]
										ABCA1↑			[148]
		↓	↓			↓		↓			↓	↓	[126]
	Tangeretin		↓	↓	CYPs↓			↑					[149]
	Chrysin								↑				[150]
					CYP2E1↑		↑						[151]
			↑					n.s.	↑		↑	↑	[130]
Isoflavonoids	Genistein							↑				↑	[152]
		↓	↓										[153]
								↑	↑				[154]
								↑					[155]
	Daidzein							↑	↑				[154]
			↑	↑									[156]

AhR: aryl hydrocarbon receptor, NRF2: nuclear factor E2-related factor 2, PXR: Pregnane X receptor, GST: glutathione S-transferases, UGT: uridine diphosphate glucuronosyltransferases, CYP1A1: cytochrome P450 family 1 subfamily A member 1, CYP1B1: cytochrome P450 family 1 subfamily B member 1, B[a]P: benzo[a]pyrene, ↑; Upregulation of gene expression, ↓; Downregulation of gene expression. n.s.: not significant.

### 3.4. B[a]P-Induced Liver Disease and Flavonoid

Chronic and acute exposure to B[a]P has been associated with an array of health issues. Recognized as a carcinogen, it significantly contributes to the development of various cancers, particularly lung and liver cancers. B[a]P exposure not only promotes processes such as cell migration, invasion, angiogenesis, and metastasis but also contributes to cancer progression [157,158]. One study reports that B[a]P induces oxidative damage in the liver and kidney of Swiss albino mice, emphasizing its carcinogenic effects [159]. Moreover, B[a]P plays a role in the pathological progression of hepatic steatosis, characterized by liver lipid accumulation, a key factor in fatty liver disease [160]. Even a singular exposure to B[a]P can induce fatty liver disease, and in combination with ethanol, it not only induces fatty liver but also exacerbates the associated damage [161]. Reports suggest that co-treatment with B[a]P and ethanol leads to oxidative stress and mitochondrial dysfunction, amplifying fatty liver disease progression [162]. The collaborative impact of B[a]P and ethanol on primary rat hepatocytes leads to plasma membrane remodeling, elevating oxidative stress, and inducing cell death [163]. Beyond its associations with cancer and fatty liver, B[a]P triggers a cascade of interconnected responses in the liver and lymphatic regions. It disrupts the morphological composition of the blood–lymph barrier, impeding lymphatic drainage in organs [164]. Despite B[a]P’s potential to induce various diseases, limited research focuses on therapeutic strategies for addressing these conditions.

The efficacy of flavonoids in mitigating diseases induced by B[a]P has been proposed. Curcumin, for instance, exhibits a protective effect against B[a]P-induced liver toxicity in rats. Its administration counteracted the toxic effects of B[a]P, restoring the normal histological architecture of the liver tissues [165]. Isoorientin, a bioactive flavonoid, displayed significant antitumor and antioxidant properties, effectively mitigating the stimulatory effect of B[a]P on ROS levels. Moreover, isoorientin demonstrated promise in alleviating B[a]P-induced febrile hepatocyte injury through the inhibition of the ROS/NF-κB/NLRP3/Caspase-1 signaling pathway [166]. This flavonoid also demonstrated positive inhibitory effects on B[a]P-induced autophagic and pyroptotic liver injury both in vitro and in vivo [167]. In a study by Hao Li et al., the adverse effects of dietary whole grains on B[a]P-induced genotoxic, oxidative, and thermogenic damage in mouse liver were validated [168]. Naringenin, another flavonoid, exhibited preventive actions against hepatocellular carcinoma by inhibiting growth factors such as TGF-β and vascular endothelial growth factor, inducing cell apoptosis, and regulating the MAPK pathway. Its selective on various proteins ensures safety in preventing liver cell carcinoma [169]. Xanthohumol is recognized as a potent inhibitor of cytochrome P450 enzymes and a stimulator of NAD(P)H:quinone reductase. One study showed that it inhibits the metabolic activation of pro-carcinogens and induces enzymes involved in carcinogen detoxification and antioxidation. Their findings substantiate the anti-genotoxic activity of xanthohumol in human cells with metabolic competence [170]. Moreover, 5,7-dimethoxyflavone demonstrated remarkable potency as an inhibitor of B[a]P-induced DNA binding and suppressed CYP1A1 protein expression and activity in Hep G2 cells. These findings suggest the potential of 5,7-dimethoxyflavone as an effective chemoprotectant against chemical-induced liver cancer [171]. 

Ongoing research delves into mitigating the toxicity induced by B[a]P through flavonoids, while concurrent auxiliary studies focus on augmenting treatment efficacy. Piperine, an adjunctive agent, amplifies the effectiveness of curcumin in alleviating B[a]P toxicity [172]. When used together, curcumin and piperine exhibit more pronounced effects than curcumin alone, notably reducing oxidative damage and chromosome aberrations induced by B[a]P. 

This comprehensive evidence not only provides insights into potential therapeutic strategies but also underscores the protective effects of flavonoids against the diverse impacts of B[a]P on liver health.

## 4. Conclusions

Following its absorption into the body, B[a]P initiates a metabolic process that generates ROS, potentially leading to diverse clinical pathologies. Flavonoids exhibit distinct antioxidant effects in countering ROS induction by B[a]P. These investigations propose a potential role for flavonoids in reducing intracellular ROS levels, thereby alleviating oxidative stress. Flavonoids demonstrate their efficacy by neutralizing ROS through interactions with free radicals and inhibiting ROS generation, highlighting their significance in countering cellular damage. In the context of B[a]P-associated research, the antioxidant properties of flavonoids serve as a natural defense mechanism against oxidative stress in hepatic cells. Moreover, flavonoids can shield against B[a]P-induced liver toxicity by regulating the xenobiotic metabolism of B[a]P. Notably, reports indicate that flavonoids have the potential to alleviate liver diseases caused by B[a]P, suggesting their role in cellular protection by preventing or minimizing B[a]P-induced liver cell damage.

## Figures and Tables

**Figure 1 antioxidants-13-00180-f001:**
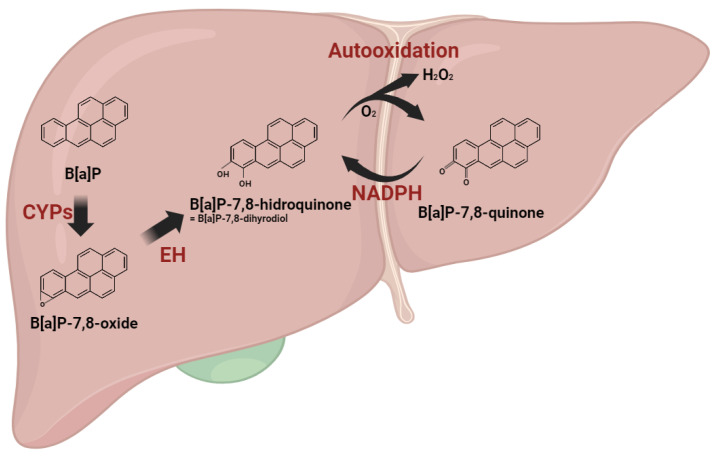
Reactive oxygen species generation during benzo[a]pyrene metabolism. B[a]P undergoes metabolic transformations initiated by cytochrome P450 enzymes (CYPs), leading to the formation of B[a]P-oxide, which is further converted into B[a]P-hydroquinone. Within the radical cation pathway, hydroxylated metabolites, including B[a]P-quinones, are generated. These B[a]P-quinones undergo enzymatic reduction facilitated by NAD(P)H quinone dehydrogenase 1 (NQO1), resulting in the formation of hydroquinone equivalents. Generated from B[a]P-quinones, hydroquinones swiftly undergo two autooxidation cycles, yielding radicals while regenerating quinones. The metabolic processing of B[a]P inevitably leads to the production of ROS.

**Figure 2 antioxidants-13-00180-f002:**
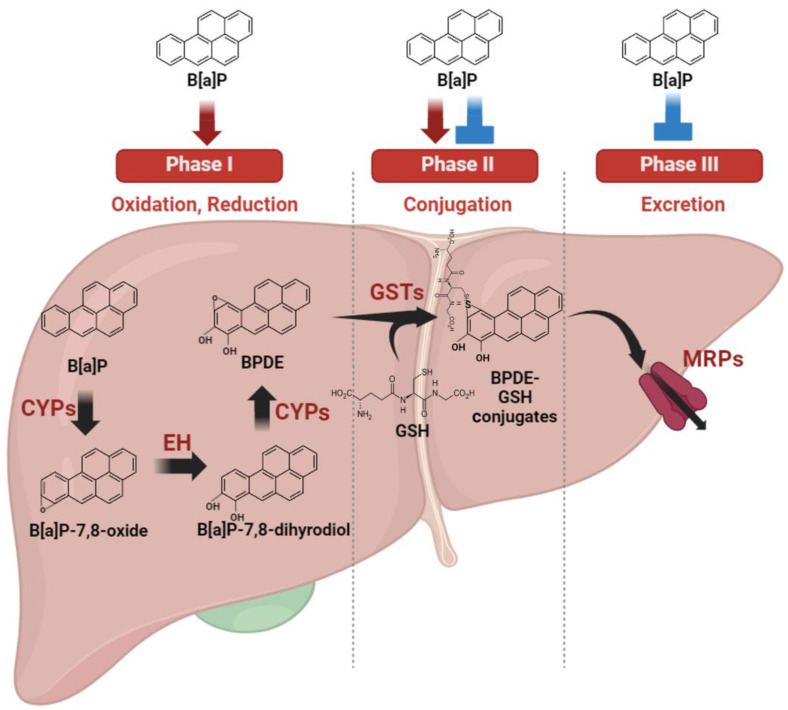
Metabolic regulation and autoregulation of benzo[a]pyrene metabolism. B[a]P undergoes a sequence of metabolic transformations catalyzed primarily, called phase I, by cytochrome P450 oxidases (CYPs) enzymes, such as CYP1A1 and CYP1B1. Phase II enzymes play a pivotal role in xenobiotic drug metabolism by converting endogenous substances and foreign compounds into more easily excretable forms. Phase III enzymes, including a diverse group of membrane transporters such as the multidrug resistance protein family (MRP), aid in the elimination of conjugated metabolites from cells. B[a]P is a substrate metabolized by these phase enzymes, and concurrently, it has the ability to regulate the expression or activity of phase enzymes.

**Figure 3 antioxidants-13-00180-f003:**
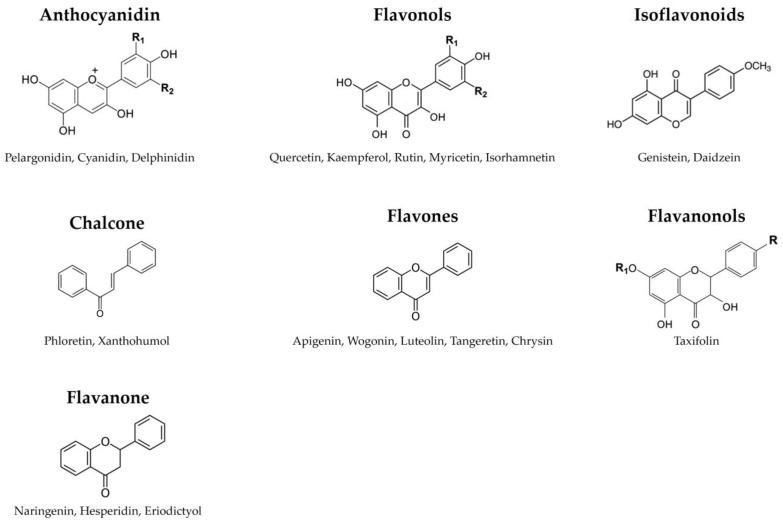
The classification of flavonoids. Flavonoids feature a 15-carbon backbone structure composed of two benzene rings linked by a heterocyclic pyran ring.

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
