# Peer review of "Hepatoprotective Effects of Flavonoids against Benzo[a]Pyrene-Induced Oxidative Liver Damage along Its Metabolic Pathways"

_antioxidants, 2024, doi:10.3390/antiox13020180_

Round 1
Reviewer 1 Report
Comments and Suggestions for Authors
This review article has summarized the current knowledge on benzo[a]pyrene (B[a]P)-induced liver injury and the role of flavonoids in alleviating oxidative stress and its toxicity.
While interesting, several questions are raised.
The editorial pinpointing out the relationship between B[a]P and flavonoids is disconcerting. A review should discuss this from a broader perspective.
The authors discuss the role of B[a]P-induced organ damage, oxidative stress, and flavonoids in this paper, but I do not understand the main message of this paper. Do the authors primarily want to communicate about B[a]P-induced organ damage and oxidative stress? Or do they primarily want to communicate the role of flavonoids in the body? If it is the former, then it is necessary to discuss not only oxidative stress, but also various other disorders and all the substances other than flavonoids that alleviate them. If it is the latter, it seems strange to limit the role of flavonoids to B[a]P-induced damage.
Reviewer 2 Report
Comments and Suggestions for Authors
The review “Hepatoprotective Effects of Flavonoids against Benzo[a]Pyrene-Induced Oxidative Liver Damages along Its Metabolic Pathways” deals on the metabolic pathways underlying Benzo[a]Pyrene toxicity to the liver, along with the main effects of flavonoids on the enzymes and metabolic regulation factors, likely helping to prevent liver damage. In general the text is well written and organized. Only minor remarks are requiring attention, as specified below.
In general, since the main topic of the review is on B[a]P, its hepatic consequences and flavonoid protection, a scheme summarizing flavonoids (only names, or better chemical structures- see for example Panche et el., J Nutr Sci.. 2016 doi: 10.1017/jns.2016.41. eCollection 2016) is to be added.
Figures 1, 2 are not recalled along the text; please name them where appropriate.
Chapter 2.2 title “Metabolic regulation by B[a]P and its hepatic consequences”, and considering for example the sentences at lines 138-139: “Phase I involves cytochrome P450 oxidases catalyzing the oxidation or reduction of xenobiotic materials. In phase II, these modified xenobiotics are conjugated with other polar compounds by specific phase II enzymes…” a title like “Metabolism and excretion of B[a]P and their hepatic consequences” seems to be more appropriate.
In Figure 2 legend, please provide extended meaning of “MRPs” abbreviation.
Tables 1,2, please give the title “Ref.s” at the head of the last column on the right.
Comments on the Quality of English Languagegood
Reviewer 3 Report
Comments and Suggestions for Authors
In this manuscript, Kim et al. tried to consolidate current knowledge on Benzo[a]pyrene B[a]P metabolism to alleviate oxidative stress and examine the role of flavonoids in mitigating its toxicity. They also intended to discover the role of flavonoids in promoting liver health and countering B[a]P-induced oxidative stress. However, this reviewer has the following concerns.
Comments:
1. Regarding Figure 1, the contents of Figure 1 are not mentioned or cited in the main text at all.
2. Similarly, Figure 2 is also not cited in the main text, although the contents are described on pages 133-210.
3. Figure legends are not found at all.
4. Regarding Table 1, antioxidant enzymes had better be classified into phase I to III, similar to table 2.
Round 2
Reviewer 1 Report
Comments and Suggestions for Authors
The authors have revised the paper appropriately based on the reviewers' input. The authors are to be commended for their efforts.
The revised version of this paper has become a very interesting review.
No further comments or corrections.
Reviewer 3 Report
Comments and Suggestions for Authors
All the comments have been addressed.